# LEARNING TO TRANSFER VIA MODELLING MULTI-LEVEL TASK DEPENDENCY

## ABSTRACT

Multi-task learning has been successful in leveraging the dependency among different tasks to improve general performance. However, when applying to 'discrete' data (graph/text), most of the existing frameworks only leverage the general task dependency with the assumption that the task dependency remains the same for (1) different data samples; and (2) different sub-structures (node/word) in one data sample (graph/text). Yet, this assumption may not be true for real-world 'discrete' datasets. Thus, we propose a novel multi-task learning framework - Learning To Transfer Via Modelling Multi-level Task Dependency, which leverages both *general task dependency* and *data-specific task dependency* to guide the knowledge transfer among tasks. We also propose a decomposition method to reduce the space complexity needed by transfer functions from quadratic to linear. To show the effectiveness of our framework and the importance of modelling multi-level task dependency, we conduct experiments on several public graph/text datasets, on which we obtain significant improvements over current frameworks.

## 1 INTRODUCTION

Multi-task learning (Caruana, 1997) aims to train a single model on multiple related tasks jointly, so that useful knowledge learned from one task can be transferred to enhance the generalization performance of other tasks. Over the last few years, different types of multi-task learning mechanisms (Sener & Koltun, 2018; Guo & Farooq, 2018; Ish, 2016; Lon, 2015) have been proposed and proved better than single-task learning methods from natural language processing (Palmer et al., 2017) and computer vision (Cortes et al., 2015) to chemical study (Ramsundar et al., 2015).

Despite the success of multi-task learning, when applying to 'discrete' data (graph/text), most of the current multi-task learning frameworks (Zamir et al., 2018; Ish, 2016) only leverage the general task dependency with the assumption that the task dependency remains the same for (1) different data samples; and (2) different sub-structures (node/word) in one data sample (graph/text). However, this assumption is not always true in many real-world problems.

(1) **Different data samples may have different task dependency.** For example, when we want to predict the chemical properties of a particular toxic molecule, despite the general task dependency, its representations learned from toxicity prediction tasks should be more significant than the other tasks.

(2) Even for the same data sample, **different sub-structures may have different task dependency.** Take sentence classification as an example. Words like 'good' or 'bad' may transfer more knowledge from sentiment analysis tasks, while words like 'because' or 'so' may transfer more from discourse relation identification tasks.

In this work, to accurately learn the task dependency in both general level and data-specific level, we propose a novel framework, 'Learning to Transfer via ModellIng mulTi-level Task dEpeNdency' (L2T-MITTEN). The *general task dependency* is learned as a parameterized weighted dependency graph. And the *data-specific task dependency* is learned with the position-wise mutual attention mechanism. The two-level task dependency can be used by our framework to improve the performance on multiple tasks. And the objective function of multi-task learning can further enhance the quality of the learned task dependency. By iteratively mutual enhancement, our framework can not

only perform better on multiple tasks, but also can extract high-quality dependency structures at different levels, which can reveal some hidden knowledge of the datasets.

Another problem is that to transfer task-specific representations between every task pair, the number of transfer functions will grow quadratically as the number of tasks increases, which is unaffordable. To solve this, we develop a universal representation space where all task-specific representations get mapped to and all target tasks can be inferred from. This decomposition method reduces the space complexity from quadratic to linear.

We validate our multi-task learning framework extensively on different tasks, including graph classification, node classification, and text classification. Our framework outperforms all the other state-of-the-art (SOTA) multi-task methods. Besides, we show that L2T-MITTEN can be used as an analytic tool to extract interpretable task dependency structures at different levels on real-world datasets.

Our contributions in this work are threefold:

- We propose a novel multi-task learning framework to learn to both general task dependency and data-specific task dependency. The learned task dependency structures can be mutually enhanced with the objective function of multi-task learning.

- We develop a decomposition method to reduce the space complexity needed by transfer functions from quadratic to linear.

- We conduct extensive experiments on different real-world datasets to show the effectiveness of our framework and the importance of modelling multi-level task dependency.

## 2 RELATED WORK

According to a recent survey (Ruder, 2017), existing multi-task learning methods can be categorized by whether they share the parameters hardly or softly.

### 2.1 HARD PARAMETER SHARING

For hard parameter sharing, a bottom network will be shared among all the tasks, and each individual task will have its own task-specific output network. The parameter sharing of the bottom network reduces the parameter needed to be learned and thus can avoid over-fitting to a specific task. However, when the tasks are not relevant enough (Sha, 2002; Baxter, 2011), the shared-bottom layers will suffer from optimization conflicts caused by mutually contradicted tasks. If the bottom model is not capable enough to encode all the necessary knowledge from different tasks, this method will fail to correctly capture all the tasks.

Besides, Dy & Krause (2018) points out that the gradients of some dominant task will be relatively larger than gradients of other tasks. This dominant phenomenon will be more obvious when the proportions of labeled data between tasks are uneven, in which case the model will be majorly optimized on data-rich tasks. To alleviate this problem, some recent works (Sener & Koltun, 2018; Dy & Krause, 2018) try to dynamically adjust the task weight during the training stage. Sener & Koltun (2018) casts the multi-task learning to a multi-objective optimization problem, and they use a gradient-based optimization method to find a Pareto optimal solution. Dy & Krause (2018) proposes a new normalization method on gradients, which attempts to balance the influences of different tasks. Recently, Guo & Farooq (2018) proposes to apply Mixture-of-Experts on multi-task learning, which linearly combines different experts (bottoms) by learnable gates. Because different experts can capture different knowledge among tasks, this model can, to some extent, model the dependency relationship among tasks.

### 2.2 SOFT PARAMETER SHARING

Methods using soft parameter sharing (Lon, 2015; Ish, 2016; Dai et al., 2015; Yan, 2017) do not keep the shared bottom layers. Instead, for soft-parameter models, most of the model parameters are task-specific. Lon (2015) focuses on reducing the annotation effort of the dependency parser tree. By combining two networks with a L2 normalization mechanism, knowledge from a different source language can be used to reduce the requirement of the amount of annotation. Further, in some

existing works, the shallow layers of the model will be separated from other layers, and be used as the feature encoders to extract task-specific representations. For example, Ish (2016) proposes a Cross-Stitch model which is a typical separate bottom model. Cross-Stitch model will be trained on different tasks separately to encode the task-specific representations from different bottom layers. Then, a cross-stitch unit is used a as a gate to combine those separately trained layers. Yan (2017) introduces the tensor factorization model to allow common knowledge to be shared at each layer in the network. By the strategy proposed in their work, parameters are softly shared across the corresponding layers of the deep learning network and the parameter sharing ratio will be determined by the model itself.

We also note that some recent works (Zamir et al., 2018; Lan et al., 2017; Liu et al., 2018) can learn to capture task dependency. Zamir et al. (2018) computes an affinity matrix among tasks based on whether the solution for one task can be sufficiently easily read out of the representation trained for another task. However, it can only capture the general task dependency. Lan et al. (2017) uses a sigmoid gated interaction module between two tasks to model their relation. But it will suffer from quadratic growth in space as the number of tasks increases. Liu et al. (2018) utilizes a shared network to share features across different tasks and uses the attention mechanism to automatically determine the importance of the shared features for the respective task. However, there is no knowledge transferred or interaction between tasks.

## 3 APPROACH

In this section, we propose our framework L2T-MITTEN, which can end-to-end learn the task dependency in both general and data-specific level, and help to assemble multi-task representations from different task-specific encoders.

### 3.1 PROBLEM FORMULATION

To formulate our framework, we first start by briefly introducing a general setting of multi-task learning. For each task $t \in \{1, ..., T\}$, we have a corresponding dataset $\{(\boldsymbol{X}_k^{(t)}, y_k^{(t)})\}_{k=1}^{N^{(t)}}$ with $N^{(t)}$ data samples, where $\boldsymbol{X}_k^{(t)}$ represent the feature vector of $k$-th data sample and $y_k^{(t)}$ is its label. We would like to train $T$ models for these tasks, and each model has its own parameter $\boldsymbol{W}^{(t)}$. Note that for different multi-task learning frameworks, these parameters can be shared hardly or softly. The goal of multi-task learning is to improve general performance by sharing information among related tasks. The total loss of multi-task learning is calculated as:

$$\min_{\boldsymbol{W}} \sum_{t=1}^{T} \frac{\sum_{k=1}^{N^{(t)}} \mathcal{L}(\boldsymbol{W}^{(t)}; \boldsymbol{X}_k^{(t)}, y_k^{(t)})}{N^{(t)}} \tag{1}$$

### 3.2 ARCHITECTURE OVERVIEW

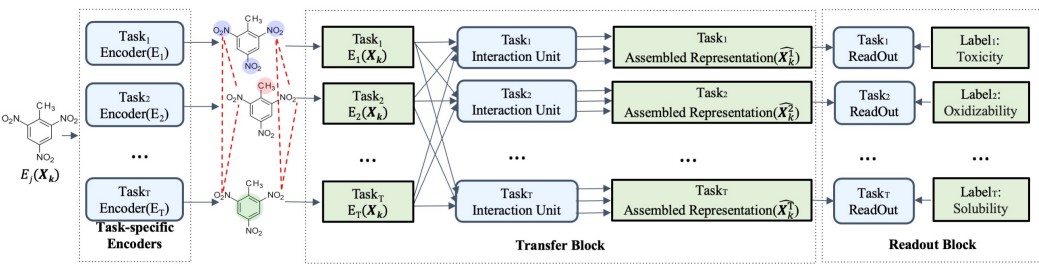

Figure 1: The overall architecture of our multi-task learning framework (L2T-MITTEN). For each input data, we will transfer its task-specific representations among different tasks, and assemble the transferred representations via a task-specific Interaction Unit to get the final representation for each task.

As is shown in Figure 1, our framework consists of three components: `Task-specific Encoders`, `Transfer Block`, and `Readout Block`.

The `Task-specific Encoders` consists of $T$ separate feature encoders, which can be any type of feed-forward networks based on specific data. Unlike hard parameter sharing methods that tie the bottom encoders' parameters together, we keep each feature encoder separate to efficiently extract task-specific knowledge. In this way, for a given data sample $\boldsymbol{X}_k$, we can use these encoders to get task-specific representations $\{E_t(\boldsymbol{X}_k)\}_{t=1}^T$, where $E_t(\boldsymbol{X}_k)$ is the representation of $\boldsymbol{X}_k$ for task $t$.

To conduct multi-task learning, one can simply use the representation of each task alone to predict labels without sharing any parameters. However, this model will suffer for tasks without sufficient labeled data. Therefore, we would like to (1) transfer the knowledge among these $T$ tasks and (2) assemble the transferred representations, which is what we do in the `Transfer Block`.

The `Readout Block` also consists of $T$ separate readout modules depending on the specific data. The detailed architecture for different tasks can be found in Appendix B.

## 3.3 TRANSFER BLOCK

### 3.3.1 TASK-SPECIFIC REPRESENTATIONS TRANSFER

In the `Transfer Block`, the first step is to transfer the task-specific representations from source to target tasks. A naive way is to use a transfer function $F_{i \to j}(\cdot)$ to transfer the task-specific representation from the space of task $i$ to task $j$ for every task pair:

$$F_{i \to j}(\boldsymbol{X}_k) = \sigma(E_i(\boldsymbol{X}_k)\boldsymbol{W}_{i \to j}), \tag{2}$$

where $\boldsymbol{W}_{i \to j} \in \mathbb{R}^{d \times d}$, and $d$ is the dimension of the task-specific representation.

However, this will result in a total number of $T^2$ transfer functions. Thus, to prevent the quadratic growth, we develop a universal representation space where all task-specific representations get mapped to and all target tasks can be inferred from. More specifically, we decompose each transfer function $F_{i \to j}(\cdot)$ to $F_{Tj} \circ F_{Si}(\cdot)$. Assume that we are trying to transfer the task-specific representation $E_i(\boldsymbol{X}_k)$ from task $i$ to task $j$, where $i, j \in \{1, 2, ..., T\}$. We can decompose the transfer matrix $\boldsymbol{W}_{i \to j}$ to $\boldsymbol{S}_i$ and $\boldsymbol{T}_j$. In this way, we only need $2T$ transfer functions in total for $F_{Si}(\cdot)$ and $F_{Tj}(\cdot)$. The space complexity is reduced from $O(T^2)$ to $O(T)$. Here we denote the transferred representation from task $i$ to task $j$ as:

$$\boldsymbol{H}_{i \to j} = F_{Tj} \circ F_{Si}(\boldsymbol{X}_i) = \sigma(E_i(\boldsymbol{X}_k)\boldsymbol{S}_i)\boldsymbol{T}_j^T, \tag{3}$$

where $\boldsymbol{S}_i, \boldsymbol{T}_j \in \mathbb{R}^{d \times d'}$ and $d'$ is a hyper-parameter.

### 3.3.2 REPRESENTATIONS ASSEMBLY VIA MULTI-LEVEL TASK DEPENDENCY

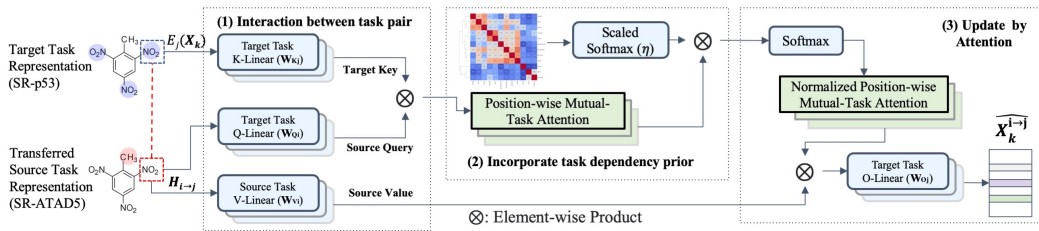

Figure 2: Position-wise mutual attention mechanism.

With the transferred representations, the next step of the `Transfer Block` is to assemble the transferred representations with respect to the multi-level task dependency. Here, the multi-level task dependency consists of two parts: (1) **the general task dependency** and (2) **the data-specific task dependency**. The multi-level task dependency is modelled by the position-wise mutual attention mechanism as shown in Figure 2.

To model the **general task dependency**, we represent it by a parameterized weighted dependency graph $\boldsymbol{D} \in \mathbb{R}^{T \times T}$. The learnable weight of this parameterized dependency graph represents the transferable weight between any task pair. Note that the dependency graph is asymmetrical. In this way, the negative influence of irrelevant tasks can be reduced as much as possible.

Further, even for the same task pair, the transferable weight (dependency) may be different for (1) different data samples; (2) different sub-structure (node/word) in one data sample (graph/text). Therefore, we study the **data-specific task dependency** in depth. To efficiently model the data-specific task dependency, we consider the mutual attention between representations of the same data sample under source and target tasks.

Given $\boldsymbol{H}_{i \to j}$, the transferred representation from the task $i$ to task $j$, and $E_j(\boldsymbol{X}_k)$, the original representation from the target task, we get the position-wise mutual attention by:

$$\boldsymbol{A}_{i \to j} = \texttt{SUM}\Big[\frac{(\boldsymbol{H}_{i \to j}\boldsymbol{W}_{Qi}) \otimes (E_j(\boldsymbol{X}_k)\boldsymbol{W}_{Kj})}{\sqrt{d''}}, \texttt{DIM=-1}\Big], \tag{4}$$

where $\boldsymbol{W}_{Qi}, \boldsymbol{W}_{Kj} \in \mathbb{R}^{d \times d''}$ are the query and key projection matrices, $d''$ is a hyper-parameter, $\otimes$ is the Hadamard product, and SUM is used to eliminate the last dimension ($d''$). We use Hadamard product instead of matrix multiplication because we only want the sub-structure in a given data sample to interact with its counterpart under other tasks. Take graph data as an example, a certain node of one graph will only give attention to the same node of that graph under other tasks.

Then, for a target task $j$, we obtain (1) a set of **general task dependency** $\boldsymbol{D}_j = \{D_{i \to j}\}_{i=1}^{T}$; and (2) a set of **data-specific task dependency** $\boldsymbol{A}_j = \{\boldsymbol{A}_{i \to j}\}_{i=1}^{T}$. To integrate them, we first scale data-specific task dependency $\boldsymbol{A}_j$ by the general task dependency $\boldsymbol{D}_j$. And then, we calculate the weighted sum of the transferred representations according to the multi-level task dependency. The final assembled representation $\hat{\boldsymbol{X}}_k^j$ (for data sample $k$ and task $j$) is as follow:

$$\hat{\boldsymbol{X}}_k^j = \sum_{i=1}^{T} \hat{\boldsymbol{X}}_k^{i \to j} = \sum_{i=1}^{T} \Big[\Big(\texttt{Softmax}(\eta \tilde{D}_{i \to j}\boldsymbol{A}_{i \to j}) \otimes (\boldsymbol{H}_{i \to j}\boldsymbol{W}_{Vi})\Big)\boldsymbol{W}_{Oj}^T\Big], \tag{5}$$

where $\tilde{\boldsymbol{D}}$ is the normalized version of $\boldsymbol{D}$, $\boldsymbol{W}_{Vi} \in \mathbb{R}^{d \times d''}$ is the value projection matrix, $\boldsymbol{W}_{Oj} \in \mathbb{R}^{d \times d''}$ is the output projection matrix. Note that $\eta$ here is a scalar parameter used to prevent the vanish gradient problem of two normalization operations.

## 4 EXPERIMENT

In this section, we evaluate the performance of our proposed L2T-MITTEN approach against several classical and SOTA approaches on two application domains: graph and text. In graph domain, we train a multitask Graph Convolutional Network (GCN) for both graph-level and node-level classification. And in text domain, we train a multitask Recurrent Neural Network (RNN) for text classification. Further, we provide visualization and analysis on the learned hidden dependency structure. Codes and datasets will be released.

### 4.1 DATASETS

For graph-level classification, we use Tox21 (Wu et al., 2017) and SIDER (Kuhn et al., 2010).

**Tox21:** Toxicology in the 21st Century (Tox21) is a database measuring the toxicity of chemical compounds. This dataset contains qualitative toxicity assays for 8014 organic molecules on 12 different targets including nuclear receptors and stress response pathways. In our experiment, we treat each molecule as a graph and each toxicity assay as a binary graph-level classification task (for 12 tasks in total).

**SIDER:** The Side Effect Resource (SIDER) is a database of marketed drugs and adverse drug reactions. This dataset contains qualitative drug side-effects measurements for 1427 drugs on 27 side-effects. In our experiment, we treat each drug (organic molecule) as a graph and the problem of predicting whether a given drug induces a side effect as a individual graph-level classification tasks (for 27 tasks in total).

For node-level classification, we use DBLP (Tang et al., 2008) and BlogCatalog (IV et al., 2009).

**DBLP:** In the dataset, authors are represented by nodes and its feature is generated by titles of their papers. Two authors are linked together if they have co-authored at least two papers in 2014-2019. We use 18 representative conferences as labels. An author is assigned to multiple labels if he/she has published papers in some conferences. The processed DBLP dataset is also published in our repository.

**BlogCatalog:** The BlogCatalog is a collection of bloggers. In the dataset, bloggers are represented as nodes, and there is a link between two bloggers if they are friends. The interests of each blogger can be tagged according to the categories that he/she published blogs in. This dataset uses 39 categories as labels. Each blogger is assigned to multiple labels if he/she published a blog in some categories.

For text classification, we use TMDb[1] dataset.

**TMDb:** The Movie Database (TMDb) dataset is a collection of information for 4803 movies. For each movie, the dataset includes information ranging from the production company, production country, release date to plot, genre, popularity, etc. In our experiment, we select plots as the input with genres as the label. We treat the problem of predicting whether a given plot belongs to a genre as an individual text-level classification tasks (for 20 tasks in total). A summary of the five datasets is provided in Appendix A.

## 4.2 BASELINE METHODS

We compare our L2T-MITTEN approach with both classical and SOTA approaches. The details are given as follows:

Single-task method:

**Single-Task:** Simply train a network consists of encoder block and readout block for each task separately.

Classical multi-task method:

**Shared-Bottom** (Caruana, 1997): This is a widely adopted multi-task learning framework which consists of a shared-bottom network (encoder block in our case) shared by all tasks, and a separate tower network (readout block in our case) for each specific task. The input is fed into the shared-bottom network, and the tower networks are built upon the output of the shared-bottom. Each tower will then produce the task-specific outputs.

SOTA multi-task methods:

**Cross-Stitch** (Ish, 2016): This method uses a "cross-stitch" unit to learn the combination of shared and task-specific representation. The "cross-stitch" unit is a $k \times k$ trainable matrix ($k$ is the number of tasks) which will transfer and fuse the representation among tasks by the following equation:

$$\begin{bmatrix} \tilde{x}_1 \\ \vdots \\ \tilde{x}_k \end{bmatrix} = \begin{bmatrix} \alpha_{11} & ... & \alpha_{1k} \\ \vdots & \ddots & \vdots \\ \alpha_{k1} & \ldots & \alpha_{kk} \end{bmatrix} \begin{bmatrix} x_1 \\ \vdots \\ x_k \end{bmatrix}$$

where $x_i$ is the output of the lower level layer for task $i$, $\alpha_{ij}$ is the transfer weight from task $j$ to task $i$, and $\tilde{x}_i$ is the input of the higher level layer for task $i$.

**MMoE** (Guo & Farooq, 2018): This method adopts the Multi-gate Mixture-of-Expert structure. This structure consists of multiple bottom networks (experts), and multiple gating networks which take the input features and output softmax gates assembling the experts with different weights. The assembled features are then passed into the task-specific tower networks.

All the baseline models use the same encoder and readout block for each task. The architecture details are provided in Appendix B.

---

[1]The Movie Database Website: https://www.themoviedb.org/.

### 4.3 Experimental Set-up

We partition the datasets into 80:20 training/testing sets (i.e. each data sample can either appear in the training or testing set) and evaluate our approach under multiple settings[2]: (1) Sufficient setting: all tasks have sufficient labeled training data; (2) Imbalanced setting: some tasks have more labeled training data than others; (3) Deficient setting: all tasks have deficient labeled training data. Models are trained for 100 epochs using the ADAM optimizer.

Table 1: Experiment result for graph datasets

| Dataset | Labeled Data Ratio | Single-Task | Shared-Bottom | Cross-Stitch | MMoE | Our |
|---|---|---|---|---|---|---|
| Tox21 | All 80% | 0.8063 | 0.8171 | 0.8204 | 0.8049 | **0.8333** |
| | Partially 10% | 0.7138 | 0.7309 | 0.7128 | 0.7331 | **0.7410** |
| | All 10% | 0.7719 | 0.7934 | 0.7823 | 0.7848 | **0.8033** |
| SIDER | All 80% | 0.6458 | 0.6484 | 0.6676 | 0.6406 | **0.6701** |
| | Partially 10% | 0.5682 | 0.5534 | 0.5504 | 0.5377 | **0.5741** |
| | All 10% | 0.6277 | 0.6290 | 0.6285 | 0.6261 | **0.6363** |
| DBLP | Partially 1% | 0.8069 | 0.8056 | 0.5148 | 0.7930 | **0.8241** |
| | All 10% | 0.8232 | 0.8077 | 0.5150 | 0.8177 | **0.8367** |
| BlogCatalog | Partially 5% | 0.5154 | 0.6521 | 0.5259 | 0.6720 | **0.6769** |
| | All 20% | 0.6100 | 0.6667 | 0.5272 | 0.6850 | **0.6861** |

Table 2: Experiment result for text dataset

| Dataset | Labeled Data Ratio | Single-Task | Shared-Bottom | Cross-Stitch | MMoE | Our |
|---|---|---|---|---|---|---|
| TMDb | All 80% | 0.8172 | 0.8272 | **0.8543** | 0.8198 | 0.8484 |
| | Partially 10% | 0.7324 | 0.7293 | 0.7234 | 0.6590 | **0.7404** |
| | All 10% | 0.8033 | 0.8227 | 0.8452 | 0.7869 | **0.8480** |

### 4.4 Result

We report the performance of our approach and baselines on graph classification, node classification and text classification tasks in terms of AUC-ROC score in Table 1 and 2 respectively.[3]

From the above result, first of all, we can see that the multi-task methods outperform the single-task method in most cases which shows the effectiveness of knowledge transfer and multi-task learning. Further, we can see that our proposed L2T-MITTEN approach outperforms both classical and SOTA in most tasks. Finally, our approach shows significant improvement under deficient labeled training data setting, since our approach leverages the structure of the data sample itself to guide the transfer among tasks.

Secondly, we found that in the real-world dataset, like DBLP dataset, our model can outperform other SOTA methods significantly, which demonstrate the importance of taking multi-level dependency into consideration. Note that the Single-Task can achieve the second-best result. This fact indicates that in the real-world dataset, tasks may be irrelevant to each other. Our multi-level task dependency can be more effective to prevent the influence of other irrelevant tasks.

Furthermore, we conduct experiments on the text classification dataset, TMDb. The Cross-Stitch model achieves the best result when the label ratio for every task is 80%. However, our task can achieve the best result for partially labeled setting (partially 10%) and few labeled setting (all 10%).

---

[2]The settings are built by applying different masks to training sets, e.g. in imbalanced setting, we randomly mask some data samples in the training set.

[3]In the tables, "All $p\%$" means all tasks' labeled training data ratios are masked to $p\%$, while "Partially $p\%$" means only some randomly selected tasks are masked to $p\%$ labeled training data.

This fact demonstrates that our directed task dependency graph can effectively prevent the negative knowledge be transferred among different tasks when the training label is few.

## 4.5 VISUALIZATION AND ANALYSIS

For visualization and analysis of the learned multi-level task dependency structure, we will take DBLP as an example here due to its simplicity in interpreting and understanding.

Figure 3: Visualization of the learned multi-level task dependency structure for DBLP dataset

(a) General tasks (conferences) dependency structure

(b) Nodes (authors) cluster

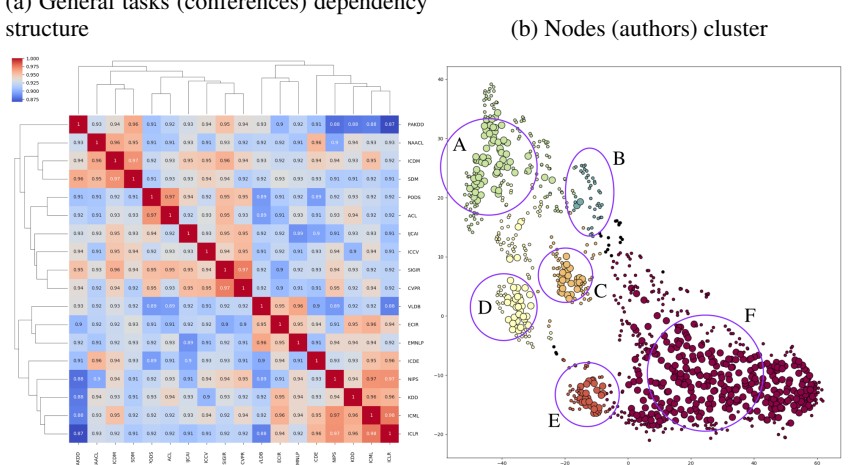

(c) Data-specific tasks (conferences) dependency structure for each cluster (Left: cluster A, Right: cluster F)

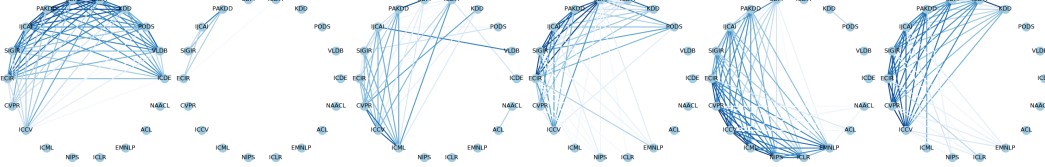

First, in Figure 3a, where we directly visualize the learned general task dependency matrix, we can see our approach indeed captures the task dependency structure in general, i.e. conferences from the same domain are more likely to be in the same sub-tree. Moreover, in Figure 3b we plot the authors (nodes) according to the learned data-specific task dependency matrix and we can see that there are some clusters formed by authors. Further, we visualize the mean value of the data-specific task dependency for each cluster, as shown in Figure 3c. We can see that different cluster does have different task dependency. This is desirable since when predicting if an author has published papers in some conferences, authors from different domains should have different transfer weight among conferences (tasks). As a summary, it is demonstrated that our approach can capture the task dependency at multiple levels according to specific data.

## 5 CONCLUSION

We propose L2T-MITTEN, a novel multi-task learning framework that (1) employs the position-wise mutual attention mechanism to learn the multi-level task dependency; (2) transfers the task-specific representations between tasks with linear space-efficiency; and (3) uses the learned multi-level task dependency to guide the inference. We design three experimental settings where training data is sufficient, imbalanced or deficient, with multiple graph/text datasets. Experimental results demonstrate the superiority of our method against both classical and SOTA baselines. We also show that our framework can be used as an analytical tool to extract the task dependency structures at different levels, which can reveal some hidden knowledge of tasks and of datasets

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

## A DATASET SUMMARY

Table 3: Graph datasets summary

| Dataset | Source | Graphs | Nodes Avg. | Edges Avg. | Graph Labels | Node Labels |
|---------|--------|--------|-----------|-----------|--------------|-------------|
| Tox21 | Bio | 8014 | 18 | 48 | 12 | - |
| SIDER | Bio | 1427 | 33 | 105 | 27 | - |
| DBLP | Citation | 1 | 14704 | 24778 | - | 18 |
| BlogCatalog | Social | 1 | 10312 | 333983 | - | 39 |

Table 4: Text dataset summary

| Dataset | Movies | Words Avg. | Genres |
|---------|--------|-----------|--------|
| TMDb | 8014 | 59 | 20 |

## B ARCHITECTURE DETAILS

### B.1 ARCHITECTURE DETAILS FOR GRAPH MODEL

As shown in Figure 4, in the Encoder Block, we use several layers of graph convolutional layers (Kipf & Welling, 2016) followed by the layer normalization (Ba et al., 2016). In the Readout Block, for graph-level task, we use set-to-set (Vinyals et al., 2015) as the global pooling operator

to extract the graph-level representation which is later fed to a classifier; while for node-level task, we simply eliminate the global pooling layer and feed the node-level representation directly to the classifier.

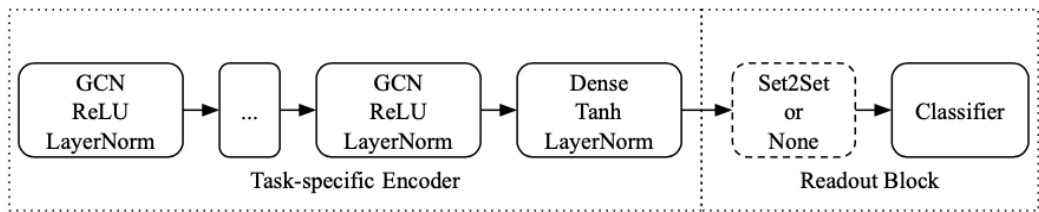

Figure 4: Graph convolutional networks architecture. Note that in node-level task, the Set2Set layer (global pooling) is eliminated.

## B.2 ARCHITECTURE DETAILS FOR TEXT MODEL

The text model uses long short-term memory (LSTM) architecture in their Encoder Block, and the dot-product attention in the Readout Block, as shown in Figure 5. The dot-product attention used to get the text-level representation is as follows:

$$\boldsymbol{\alpha} = \text{Softmax}(\boldsymbol{O}\boldsymbol{H}_n^T); \quad \hat{\boldsymbol{O}} = \boldsymbol{\alpha}^T\boldsymbol{O}$$

where $\boldsymbol{O} \in \mathbb{R}^{n \times d}$ is the output of the LSTM, $\boldsymbol{H_n} \in \mathbb{R}^{1 \times d}$ is the hidden state for the last word, $\boldsymbol{\alpha} \in \mathbb{R}^{n \times 1}$ is attention weight for each word, and $\hat{\boldsymbol{O}} \in \mathbb{R}^{1 \times d}$ is the text-level representation ($n$ is the number of words, $d$ is the feature dimension for each word).

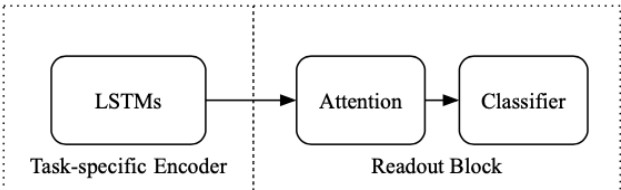

Figure 5: Text classification network architecture.

