# OpenReview forum: "Learning to Transfer via Modelling Multi-level Task Dependency"
_ICLR.cc/2020/Conference — Reject_

### Official Review · AnonReviewer2 · 2019-10-25
**Official Blind Review #2**

**Rating:** 1

**Review:**

The submission argues for the modeling the relationships between different tasks and incorporating such relationships when training multi-task frameworks. Though the basic concept (usefulness of modeling and incorporating the relationships among tasks) is valid, the submission has a number of critical issues, namely missing prior work that did that that already, missing critical specifics of the method, and unnecessary mix of different concepts.

Elaborated comments:

A) Authors seem to be unaware of critically related prior work that specially modeled task relationships and did much of what's proposed in this submission, especially "Taskonomy: Disentangling task transfer learning". The "relationship among tasks" that this submission frequently talks about is the main concept in taskonomy 2018 paper (see their abstract). Besides the apparent similarities (eg the fig 1 of this submission vs fig 1&2 of taskonomy or fig 4 of this submission vs fig 13&7 of taskonomy), the formulation has strong similarities too (transferring from "task-specific" encoders of source tasks to target tasks using transfer readout functions, or ensembling multiple task-specific representations which seem to be the same as taskonomy's higher order transfer). This submission should be majorly revised in light of prior work and the critically relevant ones should be discussed and experimentally compared to.

B) The presentation suffers from missing critical specifics. For instance, the "general task dependency" matrix  shown in Fig 3 and mentioned in page 4, which seem to be the same concept as taskonomy's task affinity matrix, is only mentioned in passing. While that seem to be one of the most important components of the method and its definition and extraction method should be discussed.

C) Inline with the point B above, the presentation of the "Transfer Block" and what the authors refer to as "Point-wise Mutual Attention Mechanism" has issues and missing details. This block could potentially have new points in it, but it's not feasible to judge that and its technical correctness given the current disposition. For instance eq 2 seem to suggest the authors develop a universal representation space where all task-specific representations get mapped to and all target tasks can be inferred from (to reduce T^2 complexity to 2T). The rest of the section does not provide a clear implementation of this and add mathematical/notation confusions. Eg H_i_j is defined to be the task-specific representation of the source task i but is indexed over both tasks i and j where j is the target, or there is a E_j(X_j) where both indexes are j while E's index is over tasks and X's index is over datapoints.

Similarly the submission seem to jump over certain concepts/terms e.g. "multi-view task dependency" in page 4 vs"multi-level task dependency" in the title, etc. What exactly "view" or "level" mean here? Are those phrases really needed? Dropping any loosely grounded phrase would be a useful practice toward a clearer presentation.

Overall, unfortunately the submission suffers from serious issues in its current shape.


----
Comments after rebuttal stage:

Thanks to the authors for the rebuttal. It provided some help, but unfortunately it doesn't resolve the majority of the issues, as most of them are too major. A clear discussion on how the proposal is different and why it is better than the recent works that were not cited would be needed, and likely authors needed strong experimental comparison with some of them, eg to prove both general and data-specific task dependency is needed.

I also didn't find the hierarchical justification clear or convincing "The reason is that different from the image which [1] focuses on, text and graph data are hierarchical: word -> sentence and node -> graph. The task dependency at the basic level (word and node) may be different from the general task dependency".


**Experience Assessment:**

I have published in this field for several years.

**Review Assessment: Checking Correctness Of Derivations And Theory:**

I carefully checked the derivations and theory.

**Review Assessment: Checking Correctness Of Experiments:**

I carefully checked the experiments.

**Review Assessment: Thoroughness In Paper Reading:**

I read the paper thoroughly.

---

> ### Author Response · Authors · 2019-11-15
> **Thank you for your constructive comments.**
>
> Thank you for your feedback. We will start by emphasizing the distinctions between our work and previous works, and then address your concerns.
>
> First, we'd like to emphasize the distinctions between our work and previous works [1][2][3].
>
> (1) Our work can capture both the general task and data-specific task dependency in “discrete” data (i.e. text and graph). The general task dependency is the same as [1]. However, for text and graph data, it is not enough to simply use the general task dependency to guide the knowledge transfer between tasks. The reason is that different from the image which [1] focuses on, text and graph data are hierarchical: word -> sentence and node -> graph. The task dependency at the basic level (word and node) may be different from the general task dependency.
>
> Take sentence classification as an example, words like “good” or “bad” may transfer more knowledge from sentiment analysis tasks, while words like “because” and “so” may transfer more from discourse relation identification task.
>
> Our work can capture the task dependency at the basic level (word and node). An extreme case would be each word/node has the same task dependency, in which our model will perform as well as [1].
>
> (2) We propose a decomposition method to reduce the size of the parameters from $O(T^2)$ to $O(T)$ (T is the number of tasks). While [2] is also capable to model the task-dependency at the word level, it suffers from quadratic complexity. [2] uses a d x d matrix $W_{sj}$ (d is the dimension of the representation for each word/node) to model the dependency between the source (s) and target task (j). However, when the number of tasks grows, the number of dependency matrix will grow quadratically ($O(T^2)$). To alleviate this, we develop a universal representation space where all task-specific representations get mapped to and all target tasks can be inferred from (eq 2).
>
> (3) Our work enables the interaction between tasks. While [3] is also able to learn the task-specific representations at different levels, there is no interaction between tasks. [3] uses a shared network to learn the task-shared representations, and T task-specific attention networks to learn the task-specific representations. However, there is no relation between tasks, and each task can only utilize the shared f representation satures from the shared network. In this case, if the tasks are not mutually strong related, [3] will suffer since the shared representations may inherently different.
>
> The aforementioned distinction guarantees that our approach has great potential to obtain better performance.
>
> Then, we will address your concerns below.
>
> Q1: “...definition and extraction method should be discussed”
>
> R1:  The general task dependency is a learnable T x T matrix (T is the number of tasks). The element at index (i, j) represents the transferable weight from task i to task j. Note that the dependency graph is asymmetrical. In this way, the negative influence of irrelevant tasks can be reduced as much as possible.
>
> Q2: “Position-wise Mutual Attention Mechanism”
>
> R2: Apologies for being unclear in this part of our paper. This part is our key contribution and we have added more details to it in the paper.
>
> We have addressed all remaining minor suggestions in the paper.
>
>
> [1] Taskonomy: Disentangling Task Transfer Learning, 2018
> [2] Multi-task Attention-based Neural Networks for Implicit Discourse Relationship Representation and Identification, 2017
> [3] End-to-End Multi-Task Learning with Attention, 2018

---

### Official Review · AnonReviewer1 · 2019-10-29
**Official Blind Review #1**

**Rating:** 3

**Review:**

This paper proposes a ‘Learning to Transfer via Modeling Multi-level Task Dependency’ for multi-task learning’, which uses the attention mechanism to learn task dependency.

In the introduction, authors claim ‘most of the current multitask learning framework rely on the assumption that all the tasks are highly correlated’. I don’t think this claim is correct. In fact, most state-of-the-art multi-task learning models can learn task dependency via different forms.

In the proposed network, different tasks have their own encoder, which leads to a large number of model parameters especially when there are a large number of tasks. This situation becomes even worse when each task has a limited number of labeled samples.

The attention has been used in multi-task learning. Authors can google ‘multi-task learning attention’ to find related works. Of course authors need to compare with those related works.

A typo: “is theposition-wise mutual attention between”

**Experience Assessment:**

I have published in this field for several years.

**Review Assessment: Checking Correctness Of Derivations And Theory:**

I assessed the sensibility of the derivations and theory.

**Review Assessment: Checking Correctness Of Experiments:**

I assessed the sensibility of the experiments.

**Review Assessment: Thoroughness In Paper Reading:**

I read the paper at least twice and used my best judgement in assessing the paper.

---

> ### Author Response · Authors · 2019-11-15
> **Thank you for your constructive comments.**
>
> Thank you for your feedback. We will start by emphasizing the distinctions between our work and previous works, and then address your concerns.
>
> First, we'd like to emphasize the distinctions between our work and previous works [1][2][3].
>
> (1) Our work can capture both the general task and data-specific task dependency in “discrete” data (i.e. text and graph). The general task dependency is the same as [1]. However, for text and graph data, it is not enough to simply use the general task dependency to guide the knowledge transfer between tasks. The reason is that different from the image which [1] focuses on, text and graph data are hierarchical: word -> sentence and node -> graph. The task dependency at the basic level (word and node) may be different from the general task dependency.
>
> Take sentence classification as an example, words like “good” or “bad” may transfer more knowledge from sentiment analysis tasks, while words like “because” and “so” may transfer more from discourse relation identification task.
>
> Our work can capture the task dependency at the basic level (word and node). An extreme case would be each word/node has the same task dependency, in which our model will perform as well as [1].
>
> (2) We propose a decomposition method to reduce the size of the parameters from $O(T^2)$ to $O(T)$ (T is the number of tasks). While [2] is also capable to model the task-dependency at the word level, it suffers from quadratic complexity. [2] uses a d x d matrix $W_{sj}$ (d is the dimension of the representation for each word/node) to model the dependency between the source (s) and target task (j). However, when the number of tasks grows, the number of dependency matrix will grow quadratically ($O(T^2)$). To alleviate this, we develop a universal representation space where all task-specific representations get mapped to and all target tasks can be inferred from (eq 2).
>
> (3) Our work enables the interaction between tasks. While [3] is also able to learn the task-specific representation at different levels, there is no interaction between tasks. [3] uses a shared network to learn the task-shared representations, and T task-specific attention networks to learn the task-specific representations. However, there is no relation between tasks, and each task can only utilize the shared representations from the shared network. In this case, if the tasks are not mutually strong related, [3] will suffer since the shared representations may inherently different.
>
> The aforementioned distinction guarantees that our approach has great potential to obtain better performance.
>
> Then, we will address your concerns below.
>
> Q1: "most state-of-the-art multi-task learning models can learn task dependency via different forms”
>
> R1: Apologies for the misunderstanding. Most state-of-the-art multi-task learning models can indeed learn task dependency via different forms. However, what we want to claim here is that our model is more robust since we can model both the general task dependency (same as several previous works) and the data-specific task dependency. We have made clarification on this in the paper.
>
> Q2: “…leads to a large number of model parameters especially when there are a large number of tasks"
>
> R2: Undeniably, our model does require a significant amount of model parameters (which are also the cases for several other multi-task learning models [1][4]. However, this does not necessarily mean that our model will suffer even when each task has a limited number of labeled samples. By modeling the multi-level task dependency (graph/text level and node/word level), our model can better utilize the task dependency information and the inner structural information from data, which increases the data efficiency. As shown in table 1&2, our model outperforms the other models under the low labeled ratio setting.
>
> Further, we are currently performing experiments on a model that uses both shared and task-specific encoder to reduce the number of parameters while maintaining the same performance. We will add the experimental results to the full version.
>
>
> [1] Taskonomy: Disentangling Task Transfer Learning, 2018
> [2] Multi-task Attention-based Neural Networks for Implicit Discourse Relationship Representation and Identification, 2017
> [3] End-to-End Multi-Task Learning with Attention, 2018
> [4] Cross-stitch Networks for Multi-task Learning, 2016

---

### Official Review · AnonReviewer4 · 2019-10-31
**Official Blind Review #4**

**Rating:** 3

**Review:**

The authors propose a multi-task learning method that uses attention mechanism to identify relations between the tasks.

Method:

- The authors motivate the use of attention mechanism for identifying a sample-dependent measure  of task relatedness by that "task dependency can be different for different data samples.." At the introduction stage this argument was not clear to me. I would suggest to expand this part of the paper by providing a stronger motivation for the proposed approach.
- what kind of mappings are used in (2)?

Experiments:

- it seems that all the datasets used are for multi-label learning. Thus, in the evaluation procedure, could the same input X appear both in the training and the test data sets (but in different tasks)? If yes, I believe it might make the evaluation less thorough. In either case it would be helpful to have this information in the description of the setting
- since use of attention is the main contribution of this work, but not the only part of the method,  I would recommend adding to the evaluation a method which is equivalent to the proposed one, but doesn't involve attention (i.e. only uses D).

Additional comments:

- in its current form the manuscript is rather hard to follow, it requires a thorough proof-reading
- it is unclear what Figure 1 on page 2 is for
- on page 2 phrase "... the label ratio is imbalanced." is confusing. I believe the authors meant that the data (not label) proportions between the tasks are uneven
- on page 3 the authors say that minimisation of the empirical risk (eq. (1)) is "the goal of multi-task learning". This sentence needs rephrasing, because from the point of view of empirical risk minimisation any multi-task approach is worse than the corresponding single-task version (i.e. its empirical risk is higher). Only in terms of the generalization performance one can argue that information sharing is beneficial.
- notation in (3) is confusing - index i is used in two meanings
- it's unclear what k in eq. (4) is
- it seems that a few references are broken

To my knowledge the idea of making amount of transfer between tasks dependent on the particular sample at hand is new. Therefore, in my opinion, with improved presentation (and in particular motivation at the beginning of the manuscript) and additional evaluation demonstrating effects of the attention component the manuscript could be recommended for acceptance.

---------------------------------------------------------------

I thank the authors for their comments. The quality fo the manuscript has indeed improved and the differences with the existing methods are clearer. However, in light of the reviewers' comments, I agree that at least the experimental section needs to be extended by adding relevant baselines. In particular, comparison to "End-to-end multi-task learning with attention" is needed to demonstrate importance of the task-level dependence measure. If direct comparison is not possible (or in addition to it) I would suggest to evaluate a modification of L2MITTEN with matrix D being equal to all 1s.

**Experience Assessment:**

I have published one or two papers in this area.

**Review Assessment: Checking Correctness Of Derivations And Theory:**

N/A

**Review Assessment: Checking Correctness Of Experiments:**

I assessed the sensibility of the experiments.

**Review Assessment: Thoroughness In Paper Reading:**

I read the paper at least twice and used my best judgement in assessing the paper.

---

> ### Author Response · Authors · 2019-11-15
> **Thank you for your constructive comments.**
>
> Thank you for acknowledging the novelty of this work and for the suggestions. Apologies for being unclear in these parts of our paper, we address your questions below.
>
> Q1a: “...stronger motivation for the proposed approach..”
>
> R1a: The motivation for the multi-level task dependency is the hierarchical structure in text and graph data (i.e. word -> sentence and node -> graph). The task dependency at word/node level may be different from the general task dependency.
>
> Take sentence classification as an example, besides the general relationship between sentiment analysis tasks and discourse relation identification tasks, words like “good” or “bad” may transfer more knowledge from sentiment analysis tasks, while words like “because” and “so” may transfer more from discourse relation identification tasks.
>
> Previous work [1] can only capture the general task dependency but does not utilize the inner hierarchical structure of “discrete” data (text and graph). An extreme case would be each word/node has the same task dependency, in which our model will perform as well as [1].
>
> Q1b: "what kind of mappings are used in (2)?”
>
> R1b: Apologies for being unclear in this part of our paper. Previously, the mapping function uses a d x d matrix $W_{ij}$ to map the representation from task i to j. However, this requires $O(T^2)$ mapping functions (T is the number of tasks). Thus, we decompose the mapping matrix $W_{ij}$ to $S_{i}T_{j}^{T}$ where $S$ and $T$ are two d x d’ matrixes. By this, the space complexity is reduced to $O(T)$. More details have been added to the paper.
>
> Q2a: "In the evaluation procedure, could the same input X appear both in the training and the test data sets (but in different tasks)?”
>
> R2a: No, an input will either be in the training set (80%) or the testing set (20%) but not both. We follow the same experimental setup as [2][3].
>
> Q2b: “...adding to the evaluation a method which is equivalent to the proposed one, but doesn't involve attention (i.e. only uses D)."
>
> R2b: Thanks for your suggestion! We added this method to the evaluation. The result shows that without attention (i.e. only uses D), the performance is similar to the cross-stitch model which pre-defined the task dependency matrix D. This is expected since both methods model the general task dependency either in a pre-defined or learnable way. We will also add this part to the full version of our paper.
>
> Q3: “Additional comments”
>
> R3: Corrected.
>
> We have re-structures the paper to improve clarity. We have also added more details in the motivation and additional evaluation in the experiment.
>
>
> [1] Taskonomy: Disentangling Task Transfer Learning, 2018
> [2] Cross-stitch Networks for Multi-task Learning, 2016
> [3] Modeling Task Relationships in Multi-task Learning with Multi-gate Mixture-of-Experts, 2018

---

### Official Review · AnonReviewer3 · 2019-10-31
**Official Blind Review #3**

**Rating:** 3

**Review:**

The paper is on an improvement of multi-task learning by considering the input tasks at two levels: (1) at task level, i.e. the relationship between the tasks and (2) by the data associated with each task. Their major argument is that most current methods hold the assumption that the tasks are correlated with each other but they conjecture that in the real-world this is not necessarily true and try to model the relationship between the input tasks at these two levels and incorporate that in the learning framework. To show effectiveness of their approach they test their method on differently oriented public datasets representing graphs, nodes and text and compare performance with some of the recent approaches to multi-task learning.

Comments to authors
1. Overall while one could get the gist of the arguments in the paper, it was not thoroughly reviewed by the authors for grammar, so it was hard to follow the finer points of the arguments. There are several grammatical mistakes and errors, on every page, it'd be too cumbersome to point them all out.
2. The distinction between the "general task dependency" and the "data dependency" does not seem significant enough. The data-dependent task dependency actually depends on the "general task dependency" as stated in the paper. This is probably manifested in the relatively slight improvement of the method compared with the SOTA. Perhaps more clarity on the difference and contribution of each "level" would make the significance stand out  clearer.

**Experience Assessment:**

I have read many papers in this area.

**Review Assessment: Checking Correctness Of Derivations And Theory:**

I did not assess the derivations or theory.

**Review Assessment: Checking Correctness Of Experiments:**

I assessed the sensibility of the experiments.

**Review Assessment: Thoroughness In Paper Reading:**

I read the paper at least twice and used my best judgement in assessing the paper.

---

> ### Author Response · Authors · 2019-11-15
> **Thank you for your constructive comments.**
>
> Thank you for your constructive comments. We address your questions as follows.
>
> Q1: “It was not thoroughly reviewed by the authors for grammar.”
>
> R1: We apologize for the grammar mistakes. We have carefully revised the paper and also re-scrutinized to improve the language.
>
> Q2: “Perhaps more clarity on the difference and contribution of each "level" would make the significance stand out clearer.”
>
> R2: The motivation for the multi-level task dependency is the hierarchical structure in text and graph data (i.e. word -> sentence and node -> graph). The task dependency at word/node level may be different from the general task dependency.
>
> Take sentence classification as an example, besides the general relationship between sentiment analysis tasks and discourse relation identification tasks, words like “good” or “bad” may transfer more knowledge from sentiment analysis tasks, while words like “because” and “so” may transfer more from discourse relation identification tasks.
>
> Previous work [1] can only capture the general task dependency but does not utilize the inner hierarchical structure of “discrete” data (text and graph). An extreme case would be each word/node has the same task dependency, in which our model will perform as well as [1].
>
>
> [1] Taskonomy: Disentangling Task Transfer Learning, 2018

---

### Decision · Program_Chairs · 2019-12-19

**Decision:**

Reject

**Comment:**

In this work, the authors address a multi-task learning setting and propose to enhance the estimation of task dependency with an attention mechanism capturing sample-dependant measure of task relatedness. All reviewers and AC agree that the current manuscript lacks clarity and convincing empirical evaluations that clearly show the benefits of the proposed approach w.r.t. state-of-the-art methods. Specifically, the reviewers raised several important concerns that were viewed by AC as critical issues:
(1) the empirical evaluations need to be significantly strengthened to show the benefits of the proposed methods over SOTA -- see R2’s request to empirically compare with the related recent work [Taskonomy, 2018] and R4’s request to compare with the work [End-to-end multi-task learning with attention, 2018]. R4 also suggested to include an ablation study to assess the benefits of the attention mechanism. Pleased to report that the authors addressed the ablation study in their rebuttal and confirmed that the proposed attention mechanism plays an important role in the performance of the proposed method.
(2) All reviewers see an issue with the presentation clarity of the conceptual and technical contributions  -- see R4’s and R2’s detailed comments and questions regarding technical contributions; see R3’s and R4’s comments that the distinction between the general task dependency and the data-driven dependency is either not significant or is not clearly articulated; finding better examples to illustrate the difference (instead of reiterating the current ones) would strengthen the clarity and conceptual contributions.
A general consensus among reviewers and AC suggests, in its current state the manuscript is not ready for a publication. It needs more clarifications, empirical studies and polish to achieve the desired goal.